

# Biochar prepared at different pyrolysis temperatures affects urea-nitrogen immobilization and N₂O emissions in paddy fields

Jiping Gao[1,*], Yanze Zhao[1,*], Wenzhong Zhang[1], Yanghui Sui[1,2], Dandan Jin[1], Wei Xin[1], Jun Yi[1] and Dawei He[1]

[1] Rice Research Institute, Liaoning Biochar Engineering & Technology Research Center, Agronomy College, Shenyang Agricultural University, Shenyang, Liaoning, China
[2] Corn Research Institute, Liaoning Academy of Agricultural Sciences, Shenyang, Liaoning, China
[*] These authors contributed equally to this work.

Corresponding authors
Wenzhong Zhang, zwzhong@126.com
Yanghui Sui, suiyanghui@126.com

## ABSTRACT

**Background**. Food safety has become a major issue, with serious environmental pollution resulting from losses of nitrogen (N) fertilizers. N is a key element for plant growth and is often one of the most important yield-limiting nutrients in paddy soil. Urea-N immobilization is an important process for restoring the levels of soil nutrient depleted by rice production and sustaining productivity. The benefits of biochar application include improved soil fertility, altered N dynamics, and reduced nutrient leaching. However, due to high variability in the quality of biochar, the responses of N loss and rice productivity to biochar amendments, especially those prepared at different pyrolysis temperatures, are still unclear. The main objectives of the present study were to examine the effects of biochar prepared at different pyrolysis temperatures on fertilizer N immobilization in paddy soil and explore the underlying mechanisms.

**Methods**. Two biochar samples were prepared by pyrolysis of maize straw at 400 °C (B400) and 700 °C (B700), respectively. The biochar was applied to paddy soil at three rates (0, 0.7, and 2.1%, w/w), with or without N fertilization (0, 168, and 210 kg N ha⁻¹). Pot experiments were performed to determine nitrous oxide (N₂O) emissions and ¹⁵N recovery from paddy soil using a ¹⁵N tracer across the rice growing season.

**Results**. Compared with the non-biochar control, biochar significantly decreased soil bulk density while increasing soil porosity, irrespective of pyrolysis temperature and N fertilizer level. Under B400 and B700, a high biochar rate decreased N loss rate to 66.42 and 68.90%, whereas a high N level increased it to 77.21 and 76.99%, respectively. Biochar also markedly decreased N₂O emissions to 1.06 (B400) and 0.75 kg ha⁻¹ (B700); low-N treatment caused a decrease in N₂O emissions under B400, but this decrease was not observed under B700. An application rate of biochar of 2.1% plus 210 kg ha⁻¹ N fertilizer substantially decreased the N fertilizer-induced N₂O emission factor under B400, whereas under B700 no significant difference was observed. Biochar combined with N fertilizer treatment decreased rice biomass and grain yield by an average of 51.55 and 23.90 g pot⁻¹, respectively, but the yield reduction under B700 was lower than under B400.

**Conclusion**. Irrespective of pyrolysis temperature, biochar had a positive effect on residual soil ¹⁵N content, while it negatively affected the ¹⁵N recovery of rice, N₂O

emissions from soil, rice biomass, and grain yield in the first year. Generally, a high application rate of biochar prepared at high or low pyrolysis temperature reduced the N fertilizer-induced $N_2O$ emission factor considerably. These biochar effects were dependent on N fertilizer level, biochar application rate, and their interactions.

# INTRODUCTION

Food safety has become a shared global concern. With the rapidly growing population, which is predicted to reach 9.8 billion by the year 2050, there is a huge demand for more food (*King et al., 2017*). Rice is the major staple in Asia, where per capita consumption is expected to increase from 84.9 kg in 2012 to 86.8 kg in 2024 (*OECD/FAO, 2015*). To increase the production of grains, fertilizer nitrogen (N) application has increased. For example, a ∼12% increase of N fertilizer level results in 11% higher grain yields of super rice varieties (*Fu & Yang, 2011*; *Liu et al., 2019*). The global average N use efficiency is 59% (*Liu et al., 2010*), indicating that nearly 40–50% of N input is lost. However, in China, the average N recovery efficiency is relatively low and ranges from 30 to 35% across the intensively cropped areas of *indica* rice (*Zhang et al., 2018*; *Yang et al., 2013*). N losses exert pressure on the environment, causing soil acidification, water pollution, and greenhouse gas (GHG) emissions (*Gruber & Galloway, 2008*). Currently, it remains challenging to meet the demands for increased food production while minimizing nitrogen-induced air and water pollution through improved N recovery (*Fixen & West, 2002*). Innovation and technologies aimed at understanding the recovery of fertilizer N in paddy systems is therefore required to provide data for higher N use efficiency (*Zhang et al., 2012b*).

Biochar application to soil is considered an effective way of mitigating the negative impacts (e.g., GHG emissions, water resource wastes, and soil degradation) of agricultural production, and the possible mechanisms involve storing carbon (C), improving soil water-holding capacity, reducing nutrient losses, and conditioning reactive N in agricultural systems (*Sun et al., 2017*; *Woolf et al., 2010*). Biochar is a solid carbon-rich product obtained via the pyrolysis conversion of biomass in an oxygen-limited environment (*International Biochar Initiative, 2012*; *Lehmann & Joseph, 2009*). Biochar, when applied as a soil amendment, can increase the fertility and quality of soil by improving soil physico-chemical properties; these effects of biochar are mainly due to its large surface area, elevated pH, high ash content, total surface charge, and high porosity (*Biederman & Harpole, 2013*). Biochar has therefore received increasing attention given its contribution to agricultural production including mitigation of GHGs emissions, soil improvement, and increased plant yield (*Gale, Sackett & Thomas, 2016*). Moreover, a significant effect of biochar on soil C storage (*Nguyen et al., 2016*) and N conversion (*Clough et al., 2013*; *Riaz et al., 2017*) is recognized. For example, *Thangarajan et al. (2018)* revealed 23 and 43% reductions in gaseous N emissions, respectively, from organic and inorganic N sources

when biochar was applied to Andisol soil. *Cayuela et al. (2014)* reported a decrease in soil $N_2O$ emissions of $49 \pm 5\%$ after field application of biochar, based on a meta-analysis involving pasture soil, hydroagric stagnic anthrosol, and loamy soil. Biochar was also found to mitigate GHG emissions at high N levels and promote nutrient uptake without fertilizer N supplementation (*Sun et al., 2017*), while aged biochar increased N use efficiency by reducing leaching or gaseous N losses in sandy soil (*Mia, Singh & Dijkstra, 2017*; *Borchard et al., 2019*).

The biochar effects depend on a number of factors, such as its intrinsic characteristics and application rate (*Cayuela et al., 2014*; *Li et al., 2019*; *Oladele, Adeyemo & Awodun, 2019*; *Restuccia et al., 2019*). Temperature and pyrolysis conditions affect biochar characteristics, having an indirect impact on soil properties, and therefore, crop growth (*Ahmad et al., 2012*; *Angın, 2013*; *Hagner et al., 2016*; *Keiluweit et al., 2010*; *Purakayastha, Kumari & Pathak, 2015*). For example, *Subedi et al. (2016)* showed that low-temperature biochar (400 °C) increased the level of soil mineral N than high-temperature biochar (600 °C) because of the more stable aromatic structure and higher hydrogen (H) and oxygen (O) contents. Higher H and O contents may increase the cation exchange capacity (e.g., ammonium ions) of soil, while a stable aromatic structure is beneficial to sequestration and transformation of soil N (*Clough et al., 2013*). However, when prepared at a temperature of 600 °C, biochar decreased the dry biomass of wheat, unlike samples prepared at 800 °C (*Tan et al., 2018*). Additionally, it was found that pyrolysis temperature determined the N release and distribution in a plant-soil system by biochar. The pyrolysis temperature of biochar therefore seems to play a crucial role in nutrient uptake by crops. With regard to GHG emissions, a review published by *Kammann et al. (2017)* shows that biochar would be effective in mitigating $N_2O$ emissions in a particular agricultural field, but the effects are still highly unpredictable. Furthermore, little is known about the effect of pyrolysis temperature and biochar application rate on urea-N fixation and $N_2O$ emissions in paddy systems.

Here a pot experiment study was carried out, with two different pyrolysis temperatures, three biochar application rates, and three N fertilizer levels. Stable isotope $^{15}N$ was used to monitor N immobilization in a rice-soil system and N uptake by rice plants. The following hypotheses were tested: (i) low pyrolysis temperature indirectly affects soil N retention for plant uptake and rice yield through increasing the surface area and lowering the pH of biochar; and (ii) higher-temperature biochar has a larger suppressive effect on soil $N_2O$ emissions because its larger pore size and higher C/N ratio resulting in lower nitrification rates compared with lower-temperature biochar.

## MATERIALS AND METHODS

### Biochar preparation and characterization

Maize straw was collected from an experimental field at Shenyang Agricultural University, Shenyang, Liaoning Province, China. After oven-drying (85 °C, 24 h), the straw was cut into small pieces (<2 mm) and stored in sealed plastic bags. The straw samples were then transferred to a rectangular porcelain container ($150 \times 100 \times 50$ mm) and placed in a

**Table 1** Properties of the two biochar samples produced by pyrolysis at temperatures of 400 °C (B400) and 700 °C (B700).

| Biochar | Total C (g kg$^{-1}$) | Total N(g kg$^{-1}$) | Surface area (m$^2$ g$^{-1}$) | Ash content(%) | Average pore size (nm) | pH(H$_2$O) | C/N |
|---------|------------------------|----------------------|-------------------------------|----------------|------------------------|------------|-----|
| B400 | 624.2 ± 6.3 | 20.0 ± 0.0 | 34.9 ± 1.2 | 14.8 ± 0.5 | 43.2 ± 0.9 | 9.8 ± 0.3 | 31.2 ± 0.3 |
| B700 | 665.5 ± 7.6 | 16.4 ± 0.7 | 12.1 ± 0.7 | 19.3 ± 0.4 | 77.7 ± 1.0 | 10.4 ± 0.4 | 40.6 ± 1.2 |

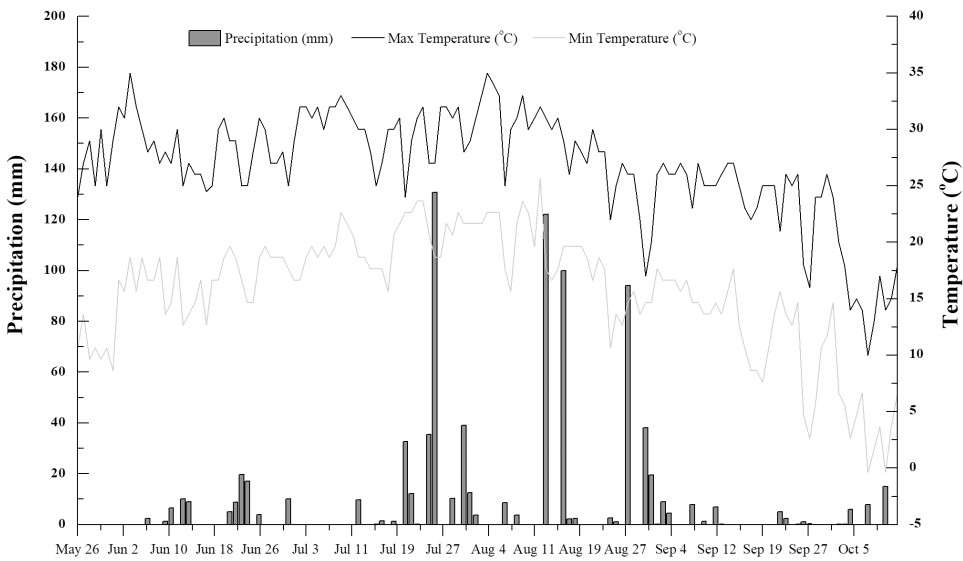

**Figure 1  Daily mean air temperature and precipitation in the study area during the rice growing season (June to October 2016).** The dark solid line represents the max temperature, light solid line represents the min temperature, histogram represents precipitation.

muffle furnace for pyrolysis at a heating rate of 15 °C min$^{-1}$. Temperature was first raised to 200 °C and then to a final temperature of 400 or 700 °C, maintained for 1 h (*Gai et al., 2014*). To minimize the oxygen content in the reaction, the container was filled with straw and tightly sealed. The biochar was cooled and then stored at room temperature until analysis and experimentation. The Brunauer-Emmett-Teller surface area and pore volume of biochar were determined using a V-Sorb 4800P surface area and porosimetry analyzer (Gold APP Instrument Corporation China, Beijing, China). Basic properties of biochar samples obtained at each different temperature are shown in Table 1. Each sample was analyzed in triplicate.

## Site description and experimental soil

Paddy soil was collected in an experimental field (41°50′N, 123°24′E) managed by the Rice Institute of Shenyang Agricultural University. The site experiences a typical semi-humid temperate continental monsoon climate, with a mean annual temperature and precipitation of 8.3 °C and 500 mm, respectively, and 183 frost-free days. The accumulated temperature (>10 °C) is 3,300–3,400 °C. Mean annual precipitation is concentrated, with slight variation in mean annual temperature (*Sui et al., 2016*). Temperature and precipitation data were recorded during the rice growing season (June to October 2016, Fig. 1).

 

The soil was taken from the surface layer (0–20 cm) in April 2016. The soil was classified as Histosols according to the United States Department of Agriculture (USDA) soil taxonomy. The basic properties of the soil prior to the experiment were as follows: initial pH = 6.8 (1:2.5, water/soil, w/v) (HANNA HI2221, Italy), bulk density = 1.46 g cm$^{-3}$, total $N = 1.87$ g kg$^{-1}$, and total $C = 15.39$ g kg$^{-1}$. The soil were air-dried, passed through a 2-mm nylon sieve, and mixed thoroughly with biochar at different rates before use.

## Experimental design

The experiment followed a $2 \times 3 \times 3$ factorial completely randomized design. There were two pyrolysis temperatures (400 and 700 °C; B400 and B700, respectively), three biochar application rates (0, 0.7, and 2.1%, w/w; equivalent to 0, 15, and 45 t ha$^{-1}$; C0, C0.7, and C2.1, respectively), and three N fertilizer levels (0, 168, and 210 kg N ha$^{-1}$; N0, N168, and N210, respectively). Each treatment group included three PVC pots (30 cm diameter $\times$ 25 cm height), giving a total of 54 pots. A total of 14 kg of soil with or without biochar was packed into each pot at a depth of 20 cm.

The rice (*Oryza sativa* L.) *japonica* variety 'Shennong 265' was cultivated in 2016. Two seedlings at the three-leaf stage were transplanted from the nursery bed to each pot on 29 May 2016. In the experiment, 36% total urea was applied before transplanting ($^{15}$N-labeled urea as base fertilizer), with 24% at the active tillering stage (unlabeled urea as tillering fertilizer) and 40% at the ear primordial stage (unlabeled urea as panicle fertilizer). The $^{15}$N-labeled urea (10.18 atom% $^{15}$N abundance) was provided by Shanghai Research Institute of Chemical Industry in Shanghai, China. In addition, all treatments received the same amounts of phosphorus (615 kg P$_2$O$_5$ ha$^{-1}$ as triple super phosphate) and potassium (200 kg K$_2$O ha$^{-1}$ as potassium chloride) as base fertilizers. Fertilizers were dissolved in water and then added into pots. All pots were watered regularly to maintain flooded conditions, with water level 5 cm above the soil surface (except for aeration at the top-tillering stage to control effective tillering). Pots were kept outdoors. To reduce the effects of precipitation, a mobile steel-framed plastic canopy (800 cm long $\times$ 300 cm wide $\times$ 200 cm high) was used.

## Plant sampling and analysis

At maturity (14 October, 2016), all plants were harvested and separated into grains and straw then oven-dried to a constant weight at 70 °C for 48 h and weighted to determine total yield. Grain moisture was determined using a hand-held moisture tester after drying (John Deere, Moline, IL, USA), and grain yield was estimated with a 14.5% moisture content.

Dry plant samples (grain and straw) were ground and sieved (0.15 mm) to analyze total N and $^{15}$N content (% in atoms) by isotope ratio mass spectrometry. $^{15}$N analyses were performed using elementar ISO prime 100 (Isoprime Ltd., Germany). Stable nuclides and the natural abundance differ from the atom% excess of the element. The background value (atom %) was subtracted from the experimental value to give the atom% excess. The natural $^{15}$N abundances of the plants and soil were estimated by averaging the values of all experimental treatments, respectively. Plant N uptake, N use efficiency, and the percentage

of plant N derived from urea fertilizer were calculated using the following equations:

$$\text{Plant N content} = \text{Total N concentration in dry biomass} \times \text{weight of dry biomass} \quad (1)$$

$$\text{Ndff} = \frac{Af\% - Acf\%}{Au\% - Acf\%} \times \text{TN(plantorsoil)} \quad (2)$$

$$\text{Ndfs} = \text{TN} - \text{Ndff} \quad (3)$$

where Ndff is the N in the plant or soil derived from $^{15}$N-labeled urea fertilizer (mg pot$^{-1}$) and Ndfs is the N in the plant derived from the soil (mg pot$^{-1}$), TN is the total N content in the plant or soil (mg pot$^{-1}$), and Au, Acf, and Af are the $^{15}$N abundance in the $^{15}$N-labeled urea fertilizer (10.18 atom%), natural $^{15}$N abundance in the plant or soil, and total $^{15}$N abundance in the plant or soil, respectively.

The recovery of $^{15}$N-labeled urea in the plant tissue or percentage retained in the soil was derived at the harvest stage using the following equation (*Bronson et al., 2000*):

$$\text{REN (\%)} = \frac{Ndff}{F} \times 100 \quad (4)$$

where $F$ is the amount of $^{15}$N-labeled urea applied (mg pot$^{-1}$).

## Soil sampling and analysis

Three soil samples were obtained using a hand-operated core sampler (inner diameter = 3.5 cm, 20 cm deep) from each pot after harvest (October 2016). Soil samples were sealed in plastic bags and maintained on ice in an insulated box then transported to the laboratory where they were stored at $-20$ °C until use. Each soil sample was divided into two parts, one for analysis of water content and another for physico-chemical analyses. Soil water content was determined after oven-drying ca. A total of 5 g of samples at 105 °C for 48 h. Before analysis of other soil properties, samples were ground using a Wiley millto and passed through a 2-mm sieve to remove root detritus. Subsamples were further passed through a 0.15-mm sieve for determination of total N using an Elementar Variomax CNS Analyzer (German Elementar Company, 2003). Soil inorganic N ($NH_4^+$-N) was extracted with 2.0 M KCl solution (*Prayogo et al., 2013*), filtered by Whatman No. 1 filter paper, and quantitated with a Continuous Flow-injection Analyzer AA3 (SEAL Analytical, Inc., Norderstedt, Germany). Bulk density was determined using a cylinder (100 cm$^3$) with additional samples collected at a 0–10 cm soil depth. Soil porosity was calculated as follows:

$$\text{TP} = (1 - \frac{\rho_b}{\rho_p}) \times 100 \quad (5)$$

$$\text{CPP} = \frac{FC}{WMC} \times BD \quad (6)$$

$$\text{AFP} = \text{TP} - \text{CPP} \quad (7)$$

where TP is the total porosity (%), $\rho_b$ is the bulk density (g cm$^{-3}$) and $\rho_p$ is the soil specific gravity (g cm$^{-3}$) in Eq. (5); CPP is the soil capillary porosity (%), FC is the field capacity (%), and WMC is the wilting moisture content (%) in Eq. (6); AFP is the soil air filled porosity (%) in Eq. (7).

## Gas sampling and analysis

$N_2O$ emission fluxes were measured across the rice growing season (June to October) using static opaque chambers (*Wang et al., 2011*). The size of the chambers (32 cm diameter × 70 or 120 cm height) was adapted to rice growth, with 70 cm height before the heading stage and 120 cm height after heading. The chambers were also wrapped in aluminum foil to reduce internal temperature changes, and equipped with circulating fans to ensure complete gas mixing during gas sampling. During the growing season, gas fluxes were measured approximately every two weeks then once more after each fertilization or water control practice. This measurement frequency was adjusted to capture the period of most active N loss in more detail.

Chambers were placed on stainless steel pedestals during the period of each flux measurement. The edge of the stainless steel pedestals had a groove filled with water to seal the gas chamber during gas collection. Gas samples were collected using a 50 mL air-tight syringe (*Singla & Inubushi, 2014*) at 15-min intervals (0, 15, 30 and 45 min) after the chambers were closed. $N_2O$ flux measurements were conducted between 8 and 10 a.m. (*Zou et al., 2005*). $N_2O$ concentrations were analyzed using a gas chromatograph (Agilent 7890A; Agilent Technologies, Santa Clara, CA, USA) and hourly emissions of $N_2O$ were determined from the slope of the mixing ratio change with four sequential samples. Quality checks were applied and $N_2O$ flux measurements were discarded if the $r^2$ of the linear regression of the fluxes was <0.90. Cumulative emissions of $N_2O$ during the growing season were calculated using trapezoidal integration to interpolate fluxes between successive sampling days (*Millar et al., 2018*).

The N fertilizer-induced $N_2O$ emission factor was calculated by the difference in cumulative $N_2O$-N emission during the rice growing season between treatments with or without N fertilization, divided by the fertilizer N applied (Eq. (8)):

$$EF\ (\%) = \frac{Cumulative\ N_2O - N\ (fertiliztion) - cumulative\ N_2O - N\ (unfertilized control)}{Fertilizer\ N\ applied} \times 100 \quad (8)$$

where EF is the N fertilizer-induced $N_2O$ emission factor.

## Statistical analysis

Treatment effects were assessed by three-way analysis of variance (ANOVA) using SPSS Statistics 18.0 (IBM, Somers, NY, USA), and significance was expressed at $P < 0.05$. All data are expressed as the mean ± standard error ($n = 3$). Multiple comparisons of means were based on Fisher's least significant difference (LSD) test at a 5% significance level unless stated otherwise.

# RESULTS

## Soil physico-chemical properties

A significant decrease in soil bulk density was observed with increasing biochar rate, irrespective of pyrolysis temperature and N level. A high biochar rate significantly increased soil porosity to 50.43–52.54% under low-temperature biochar treatment (B400) with or without N fertilization; however, under high-temperature biochar treatment (B700), this

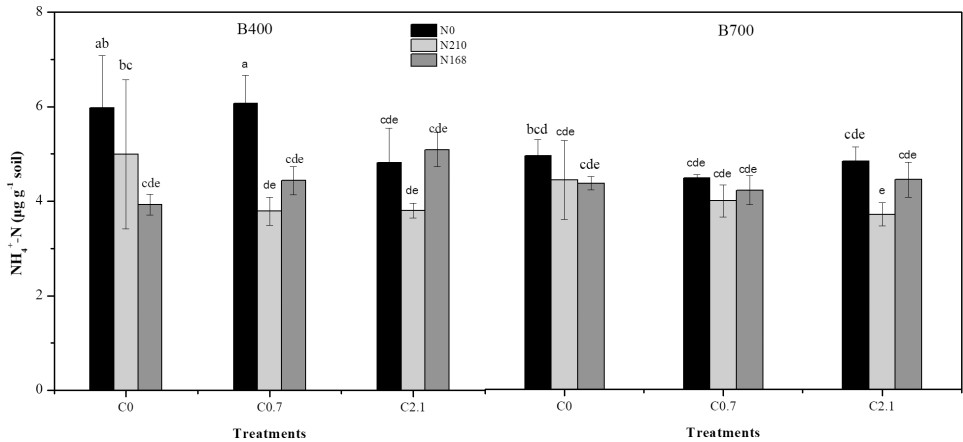

**Figure 2** **Effects of different biochar treatments with or without nitrogen fertilization on $NH_4^+$-N concentration in paddy soil.** B400 and B700 represent biochar prepared by pyrolysis at temperatures of 400 and 700 °C , respectively; C0, C0.7, and C2.1 represent biochar application rates of 0, 0.7%, and 2.1% (w/w), respectively; and N0, N210, and N168 represent urea-nitrogen fertilizer levels of 0, 168, and 210 kg N ha$^{-1}$, respectively.

increasing effect was only observed without N fertilization. Biochar had no significant effects on soil capillary porosity or air-filled porosity between the different treatments of pyrolysis temperature or biochar rate (Table 2).

Under high-N treatment (N210), soil $NH_4^+$-N decreased slightly with increasing biochar rate, irrespective of pyrolysis temperature. The lowest soil $NH_4^+$-N concentration was observed at 3.79 $\mu g\ g^{-1}$ under B400 and 3.71 $\mu g\ g^{-1}$ under B700 (Fig. 2). Biochar rate had no significant effects on soil $NH_4^+$-N.

## $^{15}$N recovery in rice-soil system

A higher biochar rate resulted in lower N uptake from $^{15}$N-labeled urea in rice, irrespective of pyrolysis temperature and N level. Under B400, a low application rate of biochar resulted in significantly higher recovery of $^{15}$N in rice compared with a high biochar rate; however, under B700, no significant difference was observed in $^{15}$N recovery between the two biochar rates. Following N fertilization alone at 210 kg ha$^{-1}$ (C0), N uptake percentage from $^{15}$N-labeled urea reached 15.80 and 15.30% under B400 and B700, respectively. Compared with non-biochar treatment, a high application rate of B400 decreased N uptake from $^{15}$N-labeled urea by 55.03 and 44.03% under N210 and N168, respectively; by contrast, B700 caused a smaller decrease in N210 (44.03%) compare with N168 (55.52%). Similarly, N uptake from the soil also decreased with increasing biochar rate (Table 3).

Following biochar application at a high rate, the recovery rate of $^{15}$N-labeled urea in rice was 7.06–7.11 and 6.78–8.56% under B400 and B700, respectively. Compared with non-biochar treatment, significant decreases were observed in the $^{15}$N recovery rate across the two pyrolysis temperatures, irrespective of N level (Table 3). The residual soil $^{15}$N rate obtained with a high biochar rate was increased compared with low-biochar and

Gao et al. (2019), *PeerJ*, DOI 10.7717/peerj.7027

**Table 2  Basic physical properties of soil under different biochar treatments with or without nitrogen fertilization.**

| Treatment[a] | | Bulk density (g cm$^{-3}$) | | | | | | Soil porosity (%) | | | | | | Capillary porosity (%) | | | | | | Air-filled porosity (%) | | | | | |
|---|---|---|---|---|---|---|---|---|---|---|---|---|---|---|---|---|---|---|---|---|---|---|---|---|---|
| | | B400 | | | B700 | | | B400 | | | B700 | | | B400 | | | B700 | | | B400 | | | B700 | | |
| | | Mean | Std. error | SL | Mean | Std. error | SL | Mean | Std. error | SL | Mean | Std. error | SL | Mean | Std. error | SL | Mean | Std. error | SL | Mean | Std. error | SL | Mean | Std. error | SL |
| N0 | C0 | 1.49 | 0.03 | a | 1.42 | 0.05 | bc | 43.83 | 1.17 | b | 46.23 | 2.03 | c | 34.73 | 1.24 | a | 34.33 | 0.68 | ab | 9.09 | 1.12 | a | 11.90 | 2.09 | bcd |
| | C0.7 | 1.42 | 0.07 | ab | 1.42 | 0.01 | bcd | 46.32 | 2.71 | ab | 46.58 | 0.34 | bc | 35.29 | 0.68 | a | **37.29** | **2.75** | **a** | 11.03 | 2.10 | a | 9.28 | 2.45 | d |
| | C2.1 | **1.26** | **0.17** | **b** | **1.27** | **0.06** | **e** | **52.54** | **6.52** | **a** | **52.05** | **2.18** | **a** | 33.17 | 2.56 | ab | 34.98 | 0.22 | ab | 19.37 | 9.05 | a | **17.07** | **1.95** | **a** |
| N210 | C0 | 1.40 | 0.10 | ab | 1.49 | 0.00 | a | 47.20 | 3.60 | ab | 43.62 | 0.15 | d | 29.43 | 2.56 | b | 30.10 | 1.53 | c | 17.77 | 6.07 | a | 13.51 | 1.58 | bc |
| | C0.7 | 1.38 | 0.02 | ab | 1.43 | 0.04 | ab | 48.09 | 0.71 | ab | 46.07 | 1.45 | c | 35.15 | 5.19 | a | 34.14 | 1.12 | b | 12.94 | 5.84 | a | 11.93 | 2.18 | bcd |
| | C2.1 | 1.31 | 0.07 | b | 1.35 | 0.04 | d | **50.43** | **2.70** | **a** | 49.09 | 1.46 | b | 34.41 | 2.70 | ab | 35.30 | 0.13 | ab | 16.02 | 4.73 | a | 13.78 | 1.54 | abc |
| N168 | C0 | 1.41 | 0.06 | ab | 1.43 | 0.03 | ab | 46.70 | 2.36 | ab | 45.96 | 1.22 | c | 32.94 | 0.69 | ab | 32.97 | 1.20 | b | 13.76 | 3.05 | a | 12.99 | 2.41 | bc |
| | C0.7 | 1.35 | 0.08 | ab | 1.38 | 0.05 | bcd | 48.95 | 3.04 | ab | 47.92 | 1.77 | bc | 32.68 | 2.78 | ab | **37.30** | **2.60** | **a** | 16.27 | 5.30 | a | 10.62 | 0.92 | cd |
| | C2.1 | **1.28** | **0.03** | **b** | 1.36 | 0.06 | cd | **51.54** | **1.11** | **a** | 48.77 | 2.17 | b | 34.37 | 1.39 | ab | 33.97 | 1.20 | b | 17.17 | 2.48 | a | 14.80 | 3.36 | ab |

Notes.

[a] B400 and B700 represent biochar prepared by pyrolysis at temperatures of 400 and 700 °C , respectively; C0, C0.7, and C2.1 represent biochar application rates of 0, 0.7%, and 2.1% (w/w), respectively; and N0, N210, and N168 represent urea-nitrogen fertilizer levels of 0, 168, and 210 kg N ha $^{-1}$, respectively. SL, significant level. Lowercase letters within each column are significantly different at $p < 0.05$.

**Table 3** Nitrogen uptake from $^{15}$N-labeled urea in rice plants under different biochar treatments at the harvest stage.

| Temperature[a] | Treatment[b] | | Ndff (mg pot$^{-1}$) | | | Ndsf (mg pot$^{-1}$) | | | REN of $^{15}$N-labeled urea in rice (%) | | |
|---|---|---|---|---|---|---|---|---|---|---|---|
| | | | Mean | Std. error | SL | Mean | Std. error | SL | Mean | Std. error | SL |
| B400 | N210 | C0 | **316.03** | **13.47** | **a** | 2,054.31 | 53.95 | a | **15.80** | **0.67** | **a** |
| | | C0.7 | 248.08 | 31.50 | b | 1,749.50 | 129.86 | ab | 12.40 | 1.57 | bc |
| | | C2.1 | 142.13 | 42.86 | de | 1,528.75 | 249.03 | bc | 7.11 | 2.14 | d |
| | N168 | C0 | 245.59 | 2.39 | b | 1,636.33 | 98.99 | bc | 15.25 | 0.15 | ab |
| | | C0.7 | 202.32 | 4.08 | bc | 1,560.99 | 25.51 | bc | 12.57 | 0.25 | bc |
| | | C2.1 | 113.63 | 18.83 | e | 1,350.94 | 158.09 | c | 7.06 | 1.17 | d |
| B700 | N210 | C0 | **306.00** | **25.56** | **a** | 2,046.47 | 81.88 | a | 15.30 | 1.28 | ab |
| | | C0.7 | 230.51 | 54.28 | b | 1,765.17 | 109.24 | ab | 11.53 | 2.71 | c |
| | | C2.1 | 171.26 | 57.10 | cd | 1,606.81 | 337.90 | bc | 8.56 | 2.86 | d |
| | N168 | C0 | 245.51 | 21.15 | b | 1,820.15 | 135.12 | ab | 15.25 | 1.31 | ab |
| | | C0.7 | 211.12 | 24.25 | bc | 1,584.70 | 143.52 | bc | 13.11 | 1.51 | abc |
| | | C2.1 | 109.21 | 21.22 | e | 1,378.55 | 216.23 | c | 6.78 | 1.32 | d |

**Notes.**
[a] B400 and B700; C0, C0.7, and C2.1; and N0, N210, and N168 are defined in Table 2 footnote.

Ndff, N content in the plant or soil derived from the $^{15}$N-labeled urea; Ndfs, N content in the plant derived from the soil; REN, recovery of $^{15}$N-labeled urea in the plant tissue; SL, Significant level (different lowercase letters in a column indicate significant difference at $p < 0.05$).

non-biochar treatments, under both B400 and B700; in particular, the difference reached a significant level under N168 ($P < 0.05$). Overall, the highest $^{15}$N recovery in the plant-soil system was obtained with biochar application under B400 (540.56 mg pot$^{-1}$) and B700 (500.77 mg pot$^{-1}$). The corresponding N recovery rates in the plant-soil system under these conditions were 33.58 and 31.10%, ranking highest among all treatments, although there were no significant differences (Table 4).

A large proportion of $^{15}$N in the rice-soil system was presumably lost, and lowest $^{15}$N loss rates were found under B400 (66.42%) and B700 (68.90%) following co-application of 2.1% biochar and 168 kg ha$^{-1}$ N fertilizer. Under B400, a smaller $^{15}$N loss rate was observed following N fertilization at 168 kg ha$^{-1}$ compared with 210 kg ha$^{-1}$, irrespective of biochar rate. No significant difference between the two pyrolysis temperatures was observed in terms of $^{15}$N loss rate. Concerning biochar rate alone, a high application rate resulted in a lower N loss rate (71.82 and 72.94%) than a low application rate (78.00 and 79.69%) under both B400 and B700. In the presence of 168 kg ha$^{-1}$ N fertilizer, a high biochar rate of B700 significantly decreased the $^{15}$N content in rice plants compared with non-biochar treatment ($P < 0.05$). Both B400 and B700 enhanced soil $^{15}$N retention at the 2.1% biochar rate, while B400 retained a larger soil $^{15}$N value compared with B700 (Table 4).

## N$_2$O emissions from paddy soil

N$_2$O emissions from the soil were significantly affected by pyrolysis temperature, biochar rate, N level, and their interactions (Table 5). N$_2$O emissions fluxes initially peaked on day 2 then decreased on day 20 after transplanting, except under co-application of 2.1% biochar and 168 kg ha$^{-1}$ N fertilizer. Under B400, co-application of 2.1% biochar and 210 kg ha$^{-1}$ N fertilizer resulted in wide fluctuations in N$_2$O emissions after water control and then a slight decrease after topdressing with N, while under B700 a sharp decrease was observed

Gao et al. (2019), *PeerJ*, DOI 10.7717/peerj.7027

**Table 4   Recovery and loss of $^{15}$N-labeled urea in the rice-soil system under different biochar treatments at the harvest stage.**

| Temperature[a] | Treatment | | Recovery of $^{15}$N in rice (mg pot$^{-1}$) | | Residual soil $^{15}$N content (mg pot$^{-1}$) | | Residual soil $^{15}$N rate (%) | | Recovery of $^{15}$N in rice-soil system (mg pot$^{-1}$) | | Recovery rate in rice-soil system (%) | | $^{15}$N lost (mg pot$^{-1}$) | | $^{15}$N loss rate (%) | |
|---|---|---|---|---|---|---|---|---|---|---|---|---|---|---|---|---|
| | | | Mean | Std. error | Mean | Std. error | Mean | Std. error | Mean | Std. error | Mean | Std. error | Mean | Std. error | Mean | Std. error |
| B400 | N210 | C0 | 316.03 | 13.47 | 169.24 | 134.51 | 8.46 | 6.73 | 485.27 | 125.44 | 24.26 | 6.27 | 1,514.73 | 125.44 | 75.74 | 6.27 |
| | | C0.7 | 248.08 | 31.50 | 159.68 | 18.06 | 7.98 | 0.90 | 407.76 | 19.31 | 20.39 | 0.97 | 1,592.24 | 19.31 | 79.61 | 0.97 |
| | | C2.1 | 142.13 | 42.86 | **313.59** | 146.55 | **15.68** | 7.33 | 455.72 | 176.95 | 22.79 | 8.85 | 1,544.28 | 176.95 | 77.21 | 8.85 |
| | N168 | C0 | 245.59 | 2.39 | 78.19 | 27.59 | 4.86 | 1.71 | 323.78 | 27.74 | 20.11 | 1.72 | 1,286.22 | 27.74 | 79.89 | 1.72 |
| | | C0.7 | 202.32 | 4.08 | 177.97 | 29.45 | 11.05 | 1.83 | 380.30 | 33.09 | 23.62 | 2.06 | 1,229.70 | 33.09 | 76.38 | 2.06 |
| | | C2.1 | 113.63 | 18.83 | **426.93** | 268.08 | **26.52** | 16.65 | 540.56 | 249.26 | **33.58** | 15.48 | 1,069.44 | 249.26 | **66.42** | 15.48 |
| B700 | N210 | C0 | 306.00 | 25.56 | 246.63 | 37.83 | 12.33 | 1.89 | 552.63 | 15.52 | 27.63 | 0.78 | 1,447.37 | 15.52 | 72.37 | 0.78 |
| | | C0.7 | 230.51 | 54.28 | 213.57 | 74.53 | 10.68 | 3.73 | 444.08 | 41.30 | 22.20 | 2.06 | 1,555.92 | 41.30 | 77.80 | 2.06 |
| | | C2.1 | 171.26 | 57.10 | 288.98 | 23.15 | 14.45 | 1.16 | 460.24 | 66.52 | 23.01 | 3.33 | 1,539.76 | 66.52 | 76.99 | 3.33 |
| | N168 | C0 | 245.51 | 21.15 | 92.85 | 26.85 | 5.77 | 1.67 | 338.36 | 31.18 | 21.02 | 1.94 | 1,271.64 | 31.18 | 78.98 | 1.94 |
| | | C0.7 | 211.12 | 24.25 | 85.53 | 27.89 | 5.31 | 1.73 | 296.64 | 52.14 | 18.43 | 3.24 | 1,313.36 | 52.14 | 81.57 | 3.24 |
| | | C2.1 | 109.21 | 21.22 | **391.56** | 97.98 | **24.32** | 6.09 | 500.77 | 118.94 | 31.10 | 7.39 | 1,109.23 | 118.94 | **68.90** | 7.39 |

**Notes.**

[a] B400 and B700

[b] C0, C0.7, and C2.1; and N0, N210, and N168 are defined in Table 2 footnote.
**Table 5  Biomass, grain yield, and $N_2O$ emission factor of rice for the treatment factors of biochar applications, pyrolysis temperature, and fertilizer and their interaction.**

| Temperature[a] | Treatment[b] | | Dry matter (g pot$^{-1}$) | | | Grain yield (g pot$^{-1}$) | | | $N_2O$ emission factor (%) | | |
|---|---|---|---|---|---|---|---|---|---|---|---|
| | | | Mean | Std. error | SL | Mean | Std. error | SL | Mean | Std. error | SL |
| B400 | N0 | C0 | 48.64 | 2.93 | c | 19.72 | 0.86 | d | | | |
| | | C0.7 | 51.84 | 3.70 | c | 22.01 | 1.79 | d | | | |
| | | C2.1 | 53.44 | 1.69 | c | 24.97 | 0.75 | d | | | |
| | N210 | C0 | 202.94 | 8.42 | a | 98.62 | 2.57 | a | 0.71 | 0.08 | a |
| | | C0.7 | 174.50 | 8.70 | ab | 86.80 | 5.47 | abc | 0.27 | 0.08 | bc |
| | | C2.1 | 159.15 | 34.24 | b | 80.34 | 11.22 | bc | 0.09 | 0.05 | e |
| | N168 | C0 | 168.49 | 9.07 | b | 85.91 | 4.25 | abc | 0.75 | 0.18 | a |
| | | C0.7 | 159.71 | 9.23 | b | 79.10 | 6.65 | bc | 0.12 | 0.10 | cde |
| | | C2.1 | 151.39 | 22.11 | b | 74.72 | 9.58 | c | 0.10 | 0.05 | e |
| B700 | N0 | C0 | 53.35 | 7.27 | c | 21.50 | 4.14 | d | | | |
| | | C0.7 | 47.04 | 0.98 | c | 19.05 | 0.17 | d | | | |
| | | C2.1 | 47.20 | 6.78 | c | 20.27 | 4.71 | d | | | |
| | N210 | C0 | 202.73 | 12.54 | a | 97.02 | 5.32 | a | 0.13 | 0.03 | cde |
| | | C0.7 | 174.02 | 1.07 | ab | 83.61 | 4.54 | abc | 0.42 | 0.12 | b |
| | | C2.1 | 173.23 | 42.64 | ab | 88.73 | 21.39 | abc | 0.12 | 0.02 | de |
| | N168 | C0 | 184.48 | 21.49 | ab | 91.01 | 12.07 | ab | 0.33 | 0.07 | b |
| | | C0.7 | 171.12 | 7.47 | ab | 84.45 | 2.77 | abc | 0.77 | 0.13 | a |
| | | C2.1 | 154.16 | 31.52 | b | 77.32 | 14.43 | bc | 0.27 | 0.05 | bcd |
| Effect | df | | F | P | | F | P | | F | P | |
| B | 2 | | 6.37 | <0.01 | | 4.57 | **0.02** | | 52.91 | <0.01 | |
| N | 2 | | 300.72 | <0.01 | | 359.58 | <0.01 | | 13.09 | <0.01 | |
| T | 1 | | 0.76 | 0.39 | | 0.28 | 0.60 | | 0.01 | 0.93 | |
| B × N | 4 | | 1.86 | 0.14 | | 1.85 | 0.14 | | 0.13 | 0.88 | |
| B × T | 2 | | 0.09 | 0.92 | | 0.11 | 0.90 | | 89.01 | <0.01 | |
| N × T | 2 | | 0.55 | 0.58 | | 0.64 | 0.53 | | 22.26 | <0.01 | |
| B × N × T | 4 | | 0.33 | 0.86 | | 0.52 | 0.72 | | 4.70 | **0.02** | |

**Notes.**
[a] B400 and B700
[b] C0, C0.7, and C2.1; and N0, N210, and N168 are defined in Table 2 footnote.
[c] Significant level: Different lowercase letters in a column indicate significant difference ($p < 0.05$).
  B, biochar application rate; N, nitrogen fertilizer level; T, pyrolysis temperature.

(Fig. 3). Cumulative $N_2O$ emissions ranged from 0.75 to 3.75 kg ha$^{-1}$ and significantly decreased with increasing biochar rate, regardless of pyrolysis temperature and fertilizer level. The application of 2.1% biochar without N fertilizer resulted in the least cumulative $N_2O$ emissions under both B400 and B700. Meanwhile, 2.1% biochar treatment caused a notable decrease in cumulative $N_2O$ emissions compared with 0.7% biochar treatment ($P < 0.05$; Fig. 4).

The N fertilizer-induced $N_2O$ emission factor, calculated as the percentage of total N supplied through urea, ranged from 0.09 to 0.77%. The lowest emission factor was obtained under B400 with 2.1% biochar plus 210 kg ha$^{-1}$ N fertilizer, while the highest was observed under B700 with 0.7% biochar plus 168 kg ha$^{-1}$ N fertilizer. The average emission factor

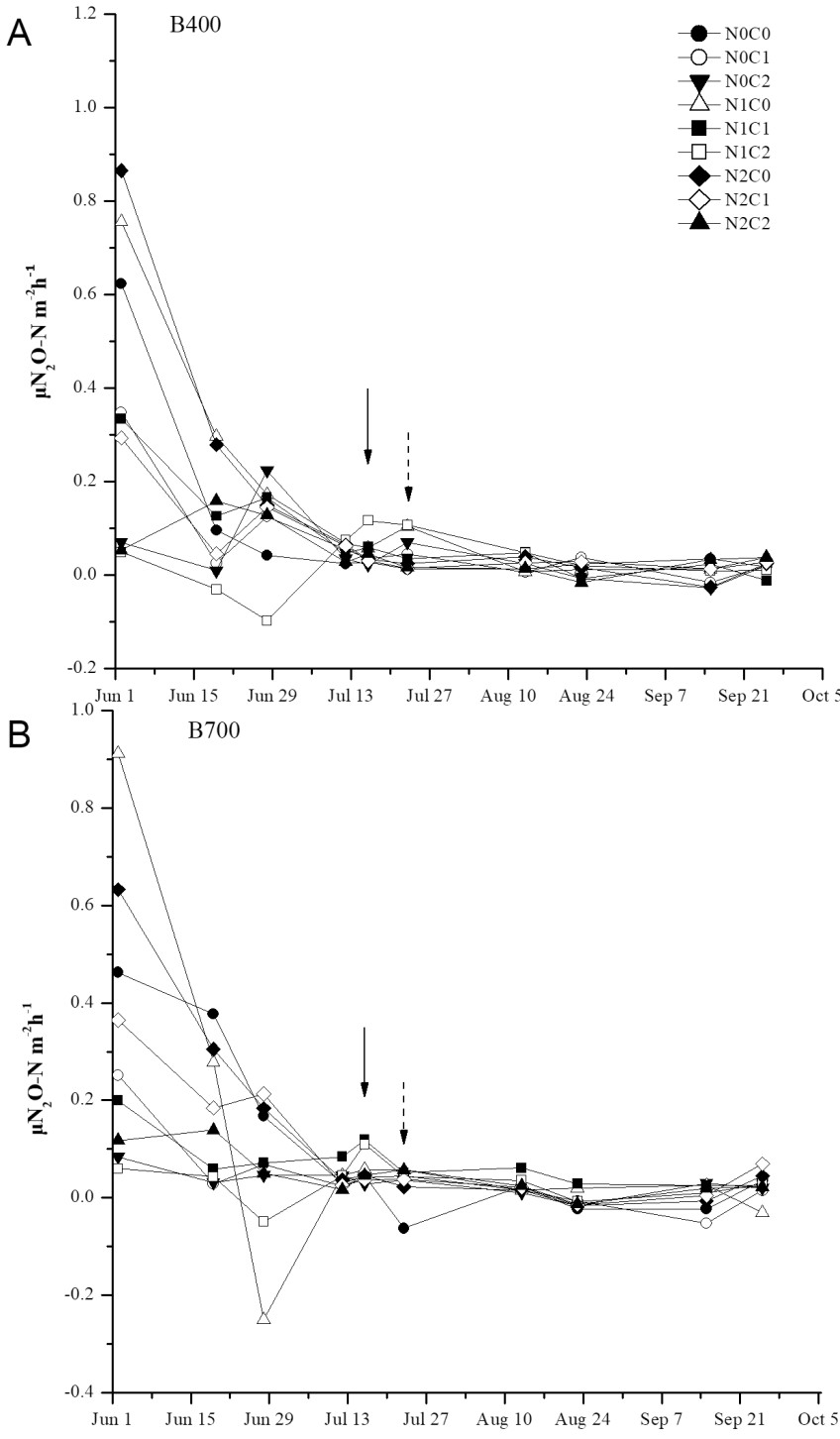

**Figure 3** **Time series of daily N₂O emissions from paddy soil under different biochar treatments during the rice growing season.** B400 and B700; C0, C0.7, and C2.1; and N0, N210, and N168 are defined in Fig. 2 caption. Solid arrows indicate water controlling, and dot arrows indicate nitrogen fertilization. Error bars represent the standard error.

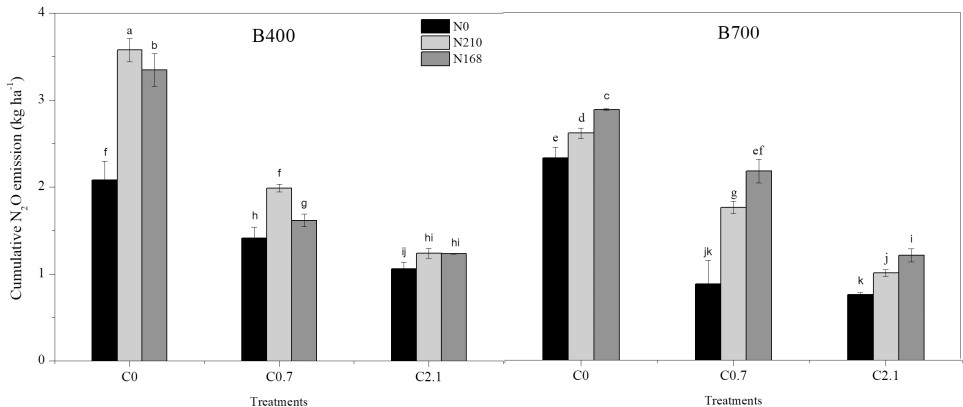

**Figure 4 Cumulative $N_2O$ emissions from paddy soil in different biochar treatments during the rice growing season.** B400 and B700; C0, C0.7, and C2.1; and N0, N210, and N168 are defined in the Fig. 2 caption.

under high-biochar treatment (C2.1) was 0.14%, which was significantly lower than that under all other biochar treatments.

### Rice biomass and yield

The aboveground biomass (including straw and grain) of rice plants at harvest was significantly higher under high-N treatment alone compared with all other treatments ($P < 0.05$). However, biochar alone had no significant effects on rice biomass, irrespective of pyrolysis temperature. Co-application of 2.1% biochar (B400) and a high N level decreased the aboveground biomass by 21.58% compared with non-biochar treatment, whereas a low N level caused a smaller decrease of 10.15%. Under B700, the rice biomass under high-N treatment alone was significantly higher than that under co-application of 2.1% biochar and 168 kg ha$^{-1}$ N fertilizer. Moreover, under 2.1% biochar (B700), high-N treatment caused an increase in biomass of 19.07 g pot$^{-1}$ compared with low-N treatment. There were no significant differences in rice biomass between biochar treatments at different pyrolysis temperatures (Table 5).

Grain yield was markedly higher under 0.7% biochar treatment compared with 2.1% biochar treatment, but only at a low pyrolysis temperature. The average yield with 0.7% biochar was 62.64 and 62.37 g pot$^{-1}$ under B400 and B700, respectively. Three-way ANOVA revealed the significant effects of biochar rate and N level on rice biomass and grain yield (Table 5).

### DISCUSSION

Different preparation conditions can alter the characteristics of biochar (*Angın, 2013*), with certain types of higher value in terms of comprehensive utilization (considering energy consumption during the pyrolysis process). In the present study, two biochar samples prepared at different pyrolysis temperatures (B400 and B700) from maize straw were used given their availability and utilization potential. Generally, B400 had ash content, average

pore size, pH, and C/N ratio, while it had a much large surface area compared with B700 (Table 1). The smaller surface area of the high-temperature biochar was likely due to the presence of large amounts of volatile organic compounds clogging biochar pores. Since biochar properties varied considerably depending on the pyrolysis temperature (Table 1), biochar effects on soil physical properties, fertilizer N immobilization, and crop growth may also differ.

## Biochar application reduces N loss in the rice-soil system

Following biochar application, fertilizer N can be lost as $NH_4^+$-N; however, the amount of $NH_4^+$-N loss is much smaller than $NO_3^-$-N (e.g., by one order of magnitude), and therefore, the $NO_3^-$-N content in paddy soil is very low and barely detectable (*Cheng et al., 2017*) . In the present study, we found that soil $NH_4^+$-N concentrations occurred at relatively low levels across treatments (3.71–6.07 $\mu$g g$^{-1}$), whereas *Riaz et al. (2017)* observed the lowest extractable $NH_4^+$-N concentration (0.51 $\mu$g g$^{-1}$) in the 2% biochar-treated soil in laboratory microcosms. Since we adopted single extractions with 2 M KCl solution, the results of soil $NH_4^+$-N might be underestimated. Interestingly, the soil $NH_4^+$-N concentration did not respond to increasing rate of biochar application alone (N0); nonetheless, a small decrease was observed with increasing biochar application following 210 kg ha$^{-1}$ N fertilization (N210), irrespective of pyrolysis temperature (Fig. 2). *Ding et al. (2010)* also observed a slight decrease in the cumulative loss of $NH_4^+$-N in the 0–20 cm surface layer of sandy silt soil after bamboo charcoal application, which was attributed to the $NH_4^+$-N sorption capacity of biochar. In our study, we observed a significantly lower soil $NH_4^+$-N concentration under high-temperature biochar treatment (B700) than low-temperature biochar treatment (B400) with 0.7% biochar application only (Fig. 2). The variation in the $NH_4^+$-N sorption ability of biochar is thought to be due to (1) difference in the ash content between biochar samples; and (2) difference in the amount of particular functional groups on the biochar surface (*Zheng et al., 2013*). In addition, biochar often have anionic sites, which can increase the ability of soil to adsorb and reserve $NH_4^+$ (*Sohi et al., 2010*).

Another pathway of N loss is gas emission (e.g., $N_2O$), which is a major problem encountered in paddy fields (*IPCC, 2014*). Throughout our study period (i.e., the rice growing season), biochar treatments caused notable decreases in cumulative $N_2O$ emissions compared with non-biochar treatment (Fig. 4). This result corroborates previous studies that biochar application to agriculture soil can efficiently mitigate $N_2O$ emissions (*Dicke et al., 2015*, *Hagemann et al., 2017*; *Sun et al., 2017*). Interestingly, there were no spikes of $N_2O$ emissions after N fertilization events in our study, which might be related to the gas sampling at wide time intervals. However, a recent study also found that corn straw-derived biochar application to alkaline clay soil did not affect $N_2O$ emissions (*Wu et al., 2018*). Consensus has yet to be reached on how and why biochar reduces $N_2O$ emissions across different plant-soil systems (*Cayuela et al., 2014*).

In our study, $N_2O$ emissions were higher under B400 than B700 treatment (Fig. 3). The possible mechanisms underpinning the higher $N_2O$ emissions after application of low-temperature biochar (400 ° C) are as follows: (1) The N immobilization and $NH_4^+$

sorption ability of B400 was enhanced, making it a better substrate for $NH_3$ oxidation and accumulation of $NO_2^-$, a majority substrate of $N_2O$ (*Clough et al., 2013*); (2) B400 contained a higher N content, which could stimulate growth of nitrifying and denitrifying bacteria, indirectly contributing to $N_2O$ emissions; (3) B400 had a lower C/N ratio, resulting in faster N cycling (*Ali, Kim & Inubushi, 2015*). *Cayuela et al. (2014)* and *Harter et al. (2016)* suggested that lower $N_2O$ emissions following biochar application were due to microbial reduction of $N_2O$ to $N_2$ via *nosZ* gene-containing microorganisms. In the current study, we did not verify *nosZ* gene abundance throughout the $N_2O$ monitoring period. Further studies are therefore needed to quantify the *nosZ* gene under different biochar treatments and thereby determine the microbiological mechanisms of $N_2O$ emission reduction.

As previously reported, biochar may alter the soil N content (*Clough et al., 2013*). However, our hypothesis that low pyrolysis temperature indirectly affects soil N retention for plant uptake was not supported by the experimental results, as no differences were found in the recovery of $^{15}N$ in rice plants between B400 and B700 treatments. *Zhou et al. (2017)* revealed that biochar prepared with mixed materials at different temperatures caused an increase in both residual soil $^{15}N$ and subsequent plant $^{15}N$ uptake in an agricultural field in Canada (*Zhou et al., 2017*). These results suggest that biochar immobilizes soil nutrients onto its surface, thereby increasing soil $^{15}N$ concentrations at higher application rates through increased porosity and surface area (*Atkinson, Fitzgerald & Hipps, 2010*; *Liang et al., 2006*).

In our study, the pyrolysis temperature of biochar did not significantly affect the recovery rate of $^{15}N$ in rice or the residual soil $^{15}N$, yet biochar application did decrease $N_2O$ emissions in paddy soil. We assume that a large portion of N loss was emitted as $N_2$ from denitrification (*Dong et al., 2015*). The relationship between residual soil $^{15}N$ and the pyrolysis temperature of biochar observed here might be explained by the fact that high-temperature biochar favors soil fungal and bacterial colonization, in turn enhancing gaseous N losses and decreasing N retention (*Gul et al., 2015*; *Nguyen et al., 2016*). Complementary experiments further suggested that applying a low rate of high-temperature biochar (>450 °C) resulted in more correlations between microbial taxa, with a large number of microorganisms appearing to influence soil N retention (*Nelissen et al., 2014*). The higher residual soil $^{15}N$ content observed under B400 treatment was therefore not simply the function of a single factor, and further analysis of long-term biochar application in the field is therefore required to determine the agricultural-environmental win-win benefits and improve crop yield. The effects of biochar on soil microbial biomass N content, microbial activity, and N fixation processes of key factors of N fixation process in various soils also need to be studied. Consequently, the use of straw-derived biochar remains challenging, requiring an accurate pyrolysis temperature and application standards. Further research is therefore needed to support the universal use of straw-derived biochar in agriculture.

## Biochar application decreases rice productivity in short-term pot experiments

In the present study, the rice biomass response to biochar varied with pyrolysis temperature, biochar application rate, and N fertilizer level, with biochar application worsening plant growth. Surprisingly, treatment with 2.1% biochar, regardless of N fertilizer level, resulted in a ca. 13.35% decrease in rice biomass under B400 compared with non-biochar treatment (Table 5). This could be attributable to the higher contents of polycyclic aromatic hydrocarbons and volatile organic compounds in biochar prepared at lower pyrolysis temperature, which may cause bio-toxic effects (*Zhu et al., 2018*). Our results are therefore contrary to those of *Zhao et al. (2014)*, who suggested that rice straw biochar applied at 9 t ha$^{-1}$ had a positive effect on rice/wheat growth. These growth improvements could be explained by bioavailable $NH_4^+$-N and $NO_3^-$-N levels after application of straw biochar (*Sui et al., 2016*; *Dong et al., 2015*). *Rajkovich (2010)* found a small increase or no change in aboveground biomass following soil application with varying rates of feedstock biochar obtained at different pyrolysis temperatures, whereas *Dao et al. (2013)* observed a three-fold increase in aboveground biomass after application of 80 t ha$^{-1}$ biochar compared with the non-biochar control. Overall, therefore, the plant biomass response to biochar depends not only on the characteristics of the biochar, the application rate and crop species, but also on the experimental set-up and original soil conditions (*Biederman & Harpole, 2013*; *Chan et al., 2008*; *Lehmann et al., 2003*).

There is a direct relationship between crop biomass and grain yield (*Sakamoto et al., 2006*). The results obtained from our pot experiment suggest that co-application of biochar with N fertilizer could markedly decrease rice yield under B400, but not B700, compared with the non-biochar control (Table 5). Our finding lends support to *Jian et al. (2018)* that laboratory-produced biochar at higher pyrolysis temperature contained lower bioavailable concentrations of polycyclic aromatic hydrocarbons and did not release volatile organic compounds at room temperatures. Wheat-straw biochar (12 t ha$^{-1}$) was previously found to have no significant effect on rice production in the first season (*Xie et al., 2013*). However, *Zhang et al. (2012a)* revealed that application of wheat straw biochar (20 t ha$^{-1}$) resulted in a 10% increase in rice yield in the first cycle and more than 9.5% increase in the second cycle. Such sustainable yield-increasing effects of biochar were also found in other experimental studies on crops. Together, these data suggest that pyrolysis temperature has only a small effect compared with biochar application rate on the short-term yield potential in paddy soil.

## CONCLUSIONS

This study confirms that appropriate application of biochar is an effective approach to reduce N loss in rice-soil systems. The results indicate that irrespective of pyrolysis temperature, a high biochar rate could enhance base fertilizer $^{15}$N retention in soil over the rice growing season, while negatively affecting urea-N uptake and biomass production in the first year. Soil bulk density and rice yield decreased after application of both low-temperature and high-temperature biochar, although $N_2O$ emissions from the paddy

soil were markedly reduced throughout the growing season. The lowest N loss rate was obtained under low-temperature treatment with application of 168 kg ha$^{-1}$ urea plus 2.1% biochar. In conclusion, application of low-temperature biochar may be an effective strategy for mitigation of N losses in paddy fields in Northeast China.

## ACKNOWLEDGEMENTS

We thank Prof. Xiaoxue Wang for her assistance in writing assistance. The authors are grateful to the anonymous reviewers for their constructive comments that led to improvements in the manuscript. Special thanks are also due to Xiaoxi Zhen, Qingge Hou, Yan Wang, Yuanle Tong, Ying Zhao, Chen Zhao, Chengcheng Peng, and Aikui Guo for assistance with the gas sampling.

### Funding

This work was supported by the Chinese Natural Sciences Foundation (31501250), the National Key R&D Program of China (2018YFD0300306), the China Postdoctoral Science Foundation (2017M611255), and the Shenyang Science and Technology Project (17-231-1-37). The funders had no role in study design, data collection and analysis, decision to publish, or preparation of the manuscript.

### Grant Disclosures

The following grant information was disclosed by the authors:
Chinese Natural Sciences Foundation: 31501250.
National Key R&D Program of China: 2018YFD0300306.
China Postdoctoral Science Foundation: 2017M611255.
Shenyang Science and Technology Project: 17-231-1-37.

### Competing Interests

The authors declare there are no competing interests.

### Author Contributions

- Jiping Gao conceived and designed the experiments, performed the experiments, analyzed the data, prepared figures and/or tables, approved the final draft.
- Yanze Zhao performed the experiments, analyzed the data, approved the final draft.
- Wenzhong Zhang conceived and designed the experiments, contributed reagents/-materials/analysis tools, authored or reviewed drafts of the paper, approved the final draft.
- Yanghui Sui conceived and designed the experiments, analyzed the data, contributed reagents/materials/analysis tools, prepared figures and/or tables, authored or reviewed drafts of the paper, approved the final draft.
- Dandan Jin analyzed the data, approved the final draft.
- Wei Xin and Jun Yi performed the experiments, analyzed the data, approved the final draft.

- Dawei He performed the experiments, approved the final draft.

## Data Availability

The raw data are available as Supplemental Files.

## Supplemental Information

Supplemental information for this article can be found online at http://dx.doi.org/10.7717/peerj.7027#supplemental-information.

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
