# Peer review of "Biochar prepared at different pyrolysis temperatures affects urea-nitrogen immobilization and N2O emissions in paddy fields"

_PeerJ, doi:10.7717/peerj.7027_

## Round 0.1 · original submission · Major Revisions

The two reviewers made some important criticisms on the manuscript. I think these criticisms are well grounded and need to be addressed prior any further considerations.

·

Basic reporting

The well written and presented paper addresses the effects of two corn straw derived biochar on nitrogen use efficiencies and gaseous loss in a paddy rice growing pot trial using various rates of urea fertilization. Given previous results of biochar amendments in rice growing and their effects on N2O emissions, it is doubtless of value to investigate the underlying mechanisms and effects of different biochar characteristics.

Experimental design

see under section 3

Validity of the findings

Despite the importance of the subject, the experiment has unfortunately three important flaws impacting the value of the study.
1) The biochar was produced in a low-tech laboratory set-up where a sealed, the biomass containing container was heated to the respective pyrolysis temperature and eventually cooled down to room temperature before opening. Due to this procedure, the biochar pores and surfaces are agglutinated and clogged with condensed pyrolysis gases. This leads to low specific surface area as well as high PAH and VOC contents of the resulting solid product. Such a biochar (which would certainly not pass IBI or EBC thresholds) cannot be considered standard biochars and are likely to cause bio-toxic effects. As the intention of the experiment was to compare the effects of a low and high temperature biochar, the biochars used in the experiment cannot be considered representative for biochars made at HTTs of 400°C, respectively, 700°C.
2) NH4 analyses of the biochar amended soils were done after single 2M KCl extractions. However, as shown in several studies (Hagemann et al., 2017; Haider et al., 2017; Kammann et al., 2015), serial extractions are necessary to correctly analyze the mineral N in biochar or biochar containing samples. The large underestimation of NH4 in the soil system directly impacts the conclusions of the study (N-retention, N-balance).
3) The authors used very high amounts of biochar (21 t/ha, 63 t/ha) which are irrelevant for any practical application in paddy rice (at costs of at least 10,000 / 30,000 USD per ha). For scientific purposes, such high application amounts might still be interesting e.g. to reveal underlying mechanisms. However, the experimental set-up did not allow any mechanistic interpretation and only showed pure effects.

Additional comments

For the above reasons, the reviewer estimates the scientific value of the manuscript as rather restrained though the execution of the pot trial and the redaction of the manuscripts follows good standards.

More comments can be found directly in the PDF version of the manuscript.

Reviewer 2 ·

Basic reporting

The article reports on a 3 way full factorial experiment rice pot experiment using 15N-urea as fertilizer. The most interesting findings of the article are that biochar can decrease both the yield and the losses as N2O.
English is generally good.
Literature is up to date, but the literature on biochar and nitrogen is very vast and I suggest the authors to review their citations to include either only review papers or studies that have been conducted in a very similar environment, for example limit the citations to biochar in rice fields. For example at some point the authors cite a study from Gonzaga 2018. Positive and negative effects of biochar from coconut husks, orange bagasse and pine wood chips on maize (Zea mays L.) growth and nutrition, to justify a decrease in biomass, I am sure you can find a study that fit better your case.
Raw data are shared but some part of the statistical tests are in chinese
The hypotheses are not very clear, specially the first one. It would be specially important that the authors describe exactely the mechanisms behind the hypotheses that they want to test.
The results are interesting, specially the one about the recovery of N in the plants and the N2O emission factor. The experiment is well conducted.

Experimental design

The research is original, the research question is well defined (effect of biochar on several environmental parameters, like emission factor, etc) though the hypotheses is not really well defined (see previous point and general comments).
The methods are clearly described, except for: biochar surface area and porosity, and soil porosity, soil capillary porosity, and soil air filled porosity.
It would have been great if the authors would have collected leached water and measured 15N in leachates. The authors define N loss all the 15N that was not recovered at the end of the study, this heavily rely on the assumption that there are no mistakes and all the 15N applied is recovered in one pool or another (excluded a little of N2O that cannot be captured). The current setup is acceptable, but if 15N recovery was estimated from all pools (including leached) the confidence in the results would be much stronger.

Validity of the findings

The data are correctly analyzed, I do disagree with their conclusion that:
"biochar application could enhance base fertilizer 15N retention in the soil over the rice growing season", because this is not the conclusion I would draw from table 4.
The data are robust and statistically sound.
The upfront delineation of clear hypotheses would be very helpful in the presentation of the results in a more orderly fashion. Sometimes authors attempt to describe the results by reporting individual interaction treatments (e.g. the B700 compared to B400 at 2.1%w and 210 Kg N ha-1), which makes it very hard to follow for the reader, who is more likely to be interested only in questions like: does biochar reduce N2O ?

Additional comments

ABSTRACT
The introduction of the abstract is a bit generic, I suggest the authors talk more specifically about the hypotheses that exist in literature on the effect of biochar on teh N cycle.
-"Low-(B400) and high-temperature (B700) biochar treatment reduced the N loss rate by up to 66.42 and 68.90%, respectively, following co-application of 2.1% biochar and 168 kg ha−1 N fertilizer.": I would try to simplify the reporting of this result, cause you have 18 possible combinations: 3rates * 2temperature * 3Nitrogen, so why report specifically about these two? You could report for example, Biochar reduced N loss by ~67%, whereas N addition increased N loss by xx%. I think for the readers it is more useful to report about the two factors tested than about the individual combinations (in a nutshell, go for the big target!).
-"Compared with the non-biochar control, 2.1% biochar plus[...]": substitute 2.1% biochar with "an addition rate of biochar of 2.1% w/w"
-"biochar had multiple effects on fertilizer N recovery": a bit vague, try to report only on the most important effects (e.g. reduced N2O emission) and report also the direction of the effect (increased, reduced...).
INTRODUCTION
"To increase the production of grains, fertilizer nitrogen (N) application has increased." : can you report a specific citation for increase in N fertilization in Rice in the region of the study?
"In China, the average is below 40%, indicating that nearly 40-50% of N input is lost": 1)The study from Vitousek reports on a specific case of corn-wheat rotation in North China, a bit far from your case (Rice) 2)These numbers are large scale and very specific, so indicate which of the two papers (Liu or Vitousek) is the source.
"However, meeting the demands for increased food production while minimizing adverse environmental impacts through improved N recovery remains a challenge (Fixen & West, 2002)." Why do you use the adverb "however", there's no contrast with your previous sentence. Also this is really broad introduction, narrow it down to the point.
"The application of biochar is considered an effective way of mitigating the negative impacts of agricultural production, improving nutrient uptake and conditioning reactive N in agricultural
systems (Sun et al., 2017; Woolf et al., 2010)." Through which mechanism can biochar achieve this? As it is now the statement is a bit broad.
"(Initiative, 2012;Lehmann & Joseph, 2009)" review format of citation.
"Biochar has therefore received increasing attention due to its contribution to agricultural practices": which agricultural practices?
"had a more positive effect on soil N than high-temperature biochar because of the more stable aromatic structure and higher hydrogen (H) and oxygen (O) contents.": 1) can you expand the link between the aromatic structure and positive effect on soil N? 2) What do you mean with positive effect on soil N? Does it increase available N? Does it decrease losses?
"Three levels of stable isotope 15N-traced fertilizer" reformulate as it is now it seems you had three treatment with different isotopic labels.
"pyrolysis temperature indirectly affects soil N retention for plant uptake and rice yield by affecting the quality of biochar": I suggest you elaborate on 1)The effect of pyrolysis temperature on biochar quality. Which quality are you considering (e.g. C concentration, C/N, ...) ? In which direction do you expect the change and what mechanism underlies this effect? 2) Retention in the soil or in the plant? Also here why? Hypotheses 2 seems more strict and falsifiable, even though also here the underlying mechanisms is not reported.
MATERIALS
2% w/w of biochar seems a lot it means approx 60 t biochar/ha. Considering how voluminous charcoal is, how feasible is this application rate?
Did you have gravel at the bottom of your pots?
"Basic properties of straw-derived biochar samples": Can you report on how the basic properties were measured, specially porosity and capillary porosity.
"The soil was classified as silt loam according to the United States Department of Agriculture (USDA) soil taxonomy." Silt-loam is a texture, not a class of soil taxonomy.
"Annual precipitation is concentrated, and the annual air temperature differs (Sui et al., 2016)": annual air temperature differs, a bit vague.
I assume that you did not measure NO3 because the soil was mostly underwater (but the first 5 cm not, so 1/5 of the soil was areated), if this is the case report it explicitely here as you do also in the discussion.
"The soil remained flooded to a depth of 5 cm except for aeration at the top-tillering stage to control effective tillering." At a first read I understood that the soil was waterlogged below the first 5 cm of soil, at a second reading I understood that the soil was completely waterlogged except for 5 cm of water. How did you maintain waterlogging? Did you add water regularly? Please report it.
RESULTS
Here the author consider loss all what is not recovered, under the assumption that there was no mistake in the analysis of the data or in the handling of the experiment this is correct. Nonetheless my trust in the results would be much stronger if the authors would have a measure of the 15N recovery that include the leached N, and ideally also of the 15N lost as N2O.
Please report the significance of the results. If the differences are not significant do not report them, unless they are very important and specify although not significant.
The results are very difficult to navigate in the way they are reported now, because they often report comparison between individual treatment, e.g. B400 decreased compared to B700 at 2.1% the 15N uptake.
I think that the reader is interested mainly in the two hypothesis that you reported and in the effect of two factors (biochar rate, pyrolysis temperature): so I would suggest reporting the significant effects highlighted in bold in table 2, otherwise it becomes nearly impossible navigate through the 3*2*3 combinations of treatment.
"reached 316.03 and 306.00 mg pot−1" Can you report it to what would be equivalent in ton/ha? Or percent of applied fertilizer?
Interestingly your N2O emissions tend to decrease over time (maybe until tillering you had less water?) and there are no spikes after N fertilization events (fig 3), I would consider commenting this in the discussion.
"The average emission factor under high-biochar treatment (C2.1) was 0.14%, which was significantly lower than that under all other biochar treatments": So I am a bit lost with the units. I think as a rule of thumb that the emission factor used by IPCC is 1% (i.e. if you apply 200Kg N ha you lose 2 Kg N-N2O, Grace 2016 http://dx.doi.org/10.1071/SR16091). In the result section you report 0.14% in table, i.e. much lower than expected. In the table 5 the results are 2 orders of magnitude lower than that (maybe those are fractions and not % as reported in the column heads?).
"Following biochar application, fertilizer N can be lost as NH4+-N": but the study from Cheng 2017 that you cite two lines below report the opposite: "The accumulate NH4 +-N in the leachate from the control treatment was 11.08 mg, which was significant higher (p > 0.05) than those in the biochar amended soil"
"however, the amount is very small compared with NO3–-N, and therefore, the nitrate content in paddy soil is very low and barely detectable (Cheng et al., 2017) .": but if the leached NO3 is high than also the soil NO3 must be high the NO3 must be sitting in soil before it was leached out, right? Also, I coudl not find any measurement of NO3 in Cheng et al. 2017.
"In our study, a lower soil NH4+-N concentration was observed under high-temperature biochar treatment (B700) than low-temperature treatment (B400) with 0.7% biochar application only (Fig. 2).": This does not seem to be significant as B400 is labelled cd and B700 cde. Am I right?
"The difference in the NH4+-N sorption ability of biochar is thought to be due to (1) differences in the ash content between biochar samples prepared at different temperatures; and (2)
differences in the amount of certain functional groups on the biochar surface (Zheng et al., 2013)." If the difference is not significant I would discuss it in this section, as it can be misleading for the reader.
"Team CW, Pachauri RK, Meyer LA. 2014" check the citation format.
"In our study, N2O emissions were higher under B400 treatment than B700 treatment (Fig. 3), suggesting that the enhanced N immobilization and NH4+ sorption ability of low-temperature (400oC) biochar increases N2O emissions (Clough et al., 2013).": Or the fact that B400 had more N content (but who knows in which form) may have an impact on the N2O emission.
"Complementation experiments": revise english.
Line 337-344: It's a bit vague, I'd consider removing it.
"(1) the high application rate of biochar, which increased soil pH and therefore resulted in a decrease in nutrient availability (Gonzaga et al., 2018)" : this could be tested if you have some soil left.
"the large surface area of biochar, which immobilized inorganic N in soil." But I don't see big differences in NH4 in soil at teh end of the season (figure 2).
"These growth improvements could be explained by an increase in soil C": I think you should focus on explaining the differences between your results and theirs, I would limit the speculation on explaining their results (also I am sure that adding biochar increased SOC also in your experiment).
"Crop biomass directly affects grain yield.": I am not sure I see the point of this statement.
"These findings suggest that our low-temperature biochar has a larger surface area and lower porosity compared with high-temperature biochar, as explained by the adsorption of organic molecules by the biochar surface, affecting soil pH and reducing rice yield." I feel this has been already discussed.
"This study confirms that[..]requires evaluation." : I would focus the conclusion on the finding from this study.
"The results suggest that biochar application could enhance base fertilizer 15N retention in the soil over the rice growing season": Judging from the standar error of the 15N loss rate of table 4. I don't think you can state that.
"the biochar effects were not proportional to pyrolysis temperature" Considering that you only had two temperatures I don't see how you can assess proportionaly or linearity in the effect of pyrolysis temperature.

---

## Round 0.2 · accepted · Accept

Your revisions addressed the issues raised by the reviewers and your rebuttal letter convinced me to accept your manuscript. Thank you for the efforts you put to improve the quality of your manuscript.

# ·

Basic reporting

The authors prepared a sophisticated and thorough rebuttal. They replied detailed to all comments and suggestions. The manuscript improved decisively.

The main and only problem of the paper remains the low and non-representative quality of the two biochars as discussed in my earlier reviewer comments. The authors describe the production process more detailed which is helpful. They refer to a publication by Jian et al. (2018) to proclaim low PAH contamination of their biochar, however, Jian et al. (2018) did not use analytical methods adapted to the biochar matrix (as described in the IBI and EBC standards as well as in the literature). However, as Jian et al. (2018) was accepted for publication in 2018, the authors may certainly justify the use of their biochar in the experiment, although the eventual influence on the presented results is important. Here, the absolute PAH content is of lesser importance than the fact that the porous structure is apparently clogged by condensed pyrolytic gases (causing very low BET surfaces for the low and the high-temperature biochars).
However, as the authors added a discussion on the biochar quality and made the reader aware that pyrolytic condensates and PAH may cause some of the observed effects, I think that the paper fulfills now the exigencies of the journal and can be recommended for publication.

Experimental design

see my first review of the paper and above section 1

Validity of the findings

see my first review of the paper and above section 1

Additional comments

see my first review of the paper and above section 1